Comparison of different sample preparation methods for platinum determination in cultured cells by graphite furnace atomic absorption spectrometry

Xiao Man 1
Huang Zaiju 1
Cai Jing 1
Jia Jinghui 2
Zhang Yuzeng 3
Dong Weihong 1
Wang Zehua zehuawang@163.net 1
1 Department of Obstetrics and Gynecology, Union Hospital, Tongji Medical College, Huazhong University of Science and Technology , Wuhan , China
2 Department of Obstetrics and Gynecology, Air Force General Hospital, PLA , Beijing , China
3 Department of Occupational and Environmental Health, School of Public Health, Tongji Medical College, Huazhong University of Science and Technology , Wuhan , China
Stoppelli Maria Patrizia
Electronic publication date: 2017 Jan 18
Publication date: 2017
Volume: 5
Electronic Location ID: e2873
Received 2016 Sep 20; Accepted 2016 Dec 6
Copyright: ©2017 Xiao et al.
Copyright year: 2017
Copyright holder: Xiao et al.
License: This is an open access article distributed under the terms of the Creative Commons Attribution License, which permits unrestricted use, distribution, reproduction and adaptation in any medium and for any purpose provided that it is properly attributed. For attribution, the original author(s), title, publication source (PeerJ) and either DOI or URL of the article must be cited.
License URL: https://creativecommons.org/licenses/by/4.0/

Keywords: Platinum, Method, Cells, GFAAS

Funding: National Natural Science Foundation of China 81302265 81272860 81472443 This study was financially supported by the National Natural Science Foundation of China (No. 81302265, 81272860, and 81472443). The funders had no role in study design, data collection and analysis, decision to publish, or preparation of the manuscript.

==============================
Background

Platinum-based agents are widely used in chemotherapy against solid tumors and insufficient intracellular drug accumulation is one of the leading causes of platinum resistance which is associated with poor survival of tumor patients. Thus, the detection of intracellular platinum is pivotal for studies aiming to overcome platinum resistance. In the present study, we aimed to establish a reliable graphite furnace atomic absorption spectrometry (GFAAS)-based assay to quantify the intracellular platinum content for cultured cells.

Methods

Several most commonly applied cell preparation methods, including 0.2% HNO3, 0.2% Triton X-100, concentrated nitric acid, RIPA combined with concentrated nitric acid and hydroxide, followed by GFAAS for platinum detection were compared in ovarian, cervical and liver cancer cell lines to obtain the optimal one, and parameters regarding linearity, accuracy, precision and sensitivity were evaluated. Influence of other metals on platinum detection and the storage conditions of samples were also determined.

Results

The treatment of cells with 0.2% HNO3 was superior to other approaches with fewer platinum loss and better repeatability. The recovery rate and precision of this method were 97.3%–103.0% and 1.4%–3.8%, respectively. The average recoveries in the presence of other metals were 95.1%–103.1%. The detection limit was 13.23 ug/L. The recovery rate of platinum remained acceptable even in cell samples stored in −20 °C or −80 °C for two months.

Discussion

After comparison, we found that 0.2% HNO3 was optimal for intracellular platinum quantification based on GFAAS, which presented values compatible with that of inductively-coupled plasma mass-spectrometry (ICP-MS), and this is partially attributed to the simplicity of this method. Moreover, the assay was proved to be accurate, sensitive, cost-effective and suitable for the research of platinum-based antitumor therapy.

Introduction

Platinum complexes are first-line chemotherapy agents for treatment of numerous solid malignancies, including tumors from ovary, testes, bladder, head and neck, cervix and lung (Lebwohl & Canetta, 1998; Rosenberg, 1985). Nevertheless, the therapeutic outcome of platinum complexes is largely impaired by drug resistance. The mechanisms of platinum-resistance mainly include reduced drug accumulation, elevated levels of glutathione and metallothionein, enhanced DNA repair capacity, and inhibition of apoptosis (Holohan et al., 2013; Zisowsky et al., 2007). It is therefore of great interest to identify a reliable assay offering accurate assessment of intracellular platinum in tumor cells.

With regard to the techniques for metals determination in a variety of sample types, atomic absorption spectrometry (AAS), high performance liquid chromatography (HPLC) and inductively coupled plasma-mass spectrometry (ICP-MS) are frequently employed in most research teams. AAS encompasses the technique of flame atomic absorption spectrometry (FAAS) and flameless atomic absorption spectrometry; the latter mainly includes graphite furnace atomic absorption spectrometry (GFAAS). With higher tolerance to inorganic and organic matrices and higher sensitivity compared with FAAS, GFAAS has been enjoying a high reputation as a mature and sensitive technique used in routine determination of metal contents (Chappuy et al., 2010; Dessuy et al., 2011; Lewen, 2011). With spectrophotometric, fluorescence, electrochemical and quenched phosphorescence detection methods, HPLC is characterized with high specificity and selectivity (Boiteau et al., 2013; Khuhawar & Arain, 2005; Santa, 2013). ICP-MS has captured increasing attention in recent years with substantially lower detection limits and advantages for multi-element analyses over GFAAS (Fukui et al., 2011; Ikeda et al., 2011; Lewen, 2011). Nevertheless, the time consuming operation (e.g., liquid-liquid back extractions) of HPLC is a disadvantage especially when a large number of samples need to be analyzed simultaneously as in toxic kinetic studies (De Baere et al., 2012). Additionally, the equipment and consumables (e.g., immunoaffinity columns and nebulizer) for HPLC and ICP-MS are more expensive than those for GFAAS (De Baere et al., 2012; Lewen, 2011). Hence, with respect to time-consumption, cost and complexity, GFAAS might represent an alternative to HPLC and ICP-MS for the determination of single element such as platinum.

Early in 1980s, Smeyers-Verbeke et al. developed the method for platinum determination in biological fluid by GFAAS, in which samples (blood, plasma, serum and urine) were lyophilized and wet ashed to destruct the organic material (Smeyers-Verbeke et al., 1981). After that, Cabrera-Vigue et al. established a method by microwave acid digestion to determine platinum in wine using standard addition method by GFAAS. They found that platinum levels in most wines were <10 ug/L (Cabrera-Vique et al., 1997). Recently, GFAAS was used for preparation and characterization of magnetic nanoparticles harboring platinum in mine samples. To achieve the best analytical performance, the pH value, loading time and flow rate of sample, concentration and flow rate of eluent were optimized by Ye et al. (2014). Nevertheless, most of the existing methodological researches on the usage of GFAAS are restricted to biological and industrial materials. The literature regarding the determination of platinum in cultured cell lines by GFAAS is quite limited, though the tumor cells cultured in vitro are widely employed in studies investigating drug resistance mechanisms. To address this issue, we aimed to identify a reliable GFAAS-based assay to determine the intracellular platinum content especially for cultured cells.

Sample preparation, for both tissues and cultured cells, counts a lot for an accurate measurement of platinum concentration. Without connective fibers, the time-consuming, hazardous lyophilization and wet ashing for tissue digestion are unnecessary. Several sample preparation procedures for cultured cells such as 0.2% and 0.6% nitric acid (HNO3) (Burger et al., 2010; Davis et al., 2012), 0.2% and 1% Triton X-100 (Burger et al., 1997; Yunos et al., 2011), concentrated nitric acid (Buss et al., 2011; Kalayda, Wagner & Jaehde, 2012; Rotte et al., 2010; Zisowsky et al., 2007) and hydroxide (Noordhuis et al., 2008) have been used for platinum analysis using flameless AAS (graphite tube were used) and GFAAS. In addition, another cell processing procedure (a lysis buffer + concentrated nitric acid) followed by HPLC on line with ICP-MS was also used for platinum determination (Federici et al., 2014). Nevertheless, the lack of a systematic methodological quality assessment of these assays made it blind for researchers to select an appropriate method to measure platinum concentration in cells. We therefore compared the most widely used sample processing procedures to obtain an optimal GFAAS-based assay with simplicity, veracity, and sensitivity for determination of platinum concentration in cultured cells.

Materials & Methods

Reagents

Analytical grade reagents were used exclusively. Cisplatin and Triton X-100 were respectively purchased from Qilu pharmaceutical company (China) and Amresco (America). Concentrated nitric acid (10014518), standard solutions of Ca (40272064), Mg (53203671), Zn (53205273), Cu (53205181), K (40243861), Na (40262163), Mn (53204676) and Fe (40940260) were obtained from Sinopharm Chemical Reagent Co., Ltd, Shanghai, China. Different concentrations of nitric acid and Triton X-100 and a standard solution of 300 ug/L platinum were prepared with deionized water, which was obtained from the SNW ultra-pure water system with a resistivity of 18.2 M (Heal Force, Shanghai, China).

Cell lines and cell culture

Human epithelial ovarian cancer A2780 (ECACC) and CAOV3 (ATCC) cells, cervical cancer HeLa (ATCC) cells and liver carcinoma Hep G2 (ATCC) cells were cultured in RPMI-1640 medium (Gibco, USA) supplemented with 10% fetal bovine serum (Gibco, Billings, MT, USA) in a humidified atmosphere containing 5% CO2 at 37 °C. Cisplatin-resistant variant of A2780 (A2780/CDDP) was generated by exposure to increasing concentration of cisplatin. After being incubated with cisplatin, cells were washed 3 times with ice-cold phosphate buffered saline (PBS) to rinse the dead cells and remaining drug. Then cells were harvested with trypsin and washed twice with ice-cold PBS. Ultimately, cell suspensions were divided into equal aliquots and kept frozen (−20 °C or −80 °C) after the supernatants were discarded. One of the aliquots was removed for protein content measurement via Bradford method (Beyotime Biotechnology, Haimen, Jiangsu, China), the others for platinum determination. The intracellular platinum levels were expressed as ng of Pt per aliquot or ng of Pt per mg of protein. The present study was approved by the ethical committee of Union Hospital, Tongji Medical College, Huazhong University of Science and Technology, China (Approval number:  2014073).

Sample preparation and operating condition of instrument

All cell aliquots were kept frozen (−20 °C) until the day for GFAAS analysis. Immediately after thawing, cell pellets were respectively processed according to following procedures: ① 0.2% HNO3 at room temperature (Burger et al., 2010), ② 0.2% Triton X-100 on ice (Burger et al., 1997), ③ concentrated nitric acid at 60 °C for 20 min (Rotte et al., 2010; Zisowsky et al., 2007), ④ concentrated nitric acid at 80 °C for 1 h (Buss et al., 2011; Kalayda, Wagner & Jaehde, 2012), ⑤ RIPA (Beyotime Biotechnology, Haimen, Jiangsu, China) complemented with concentrated nitric acid at 60 °C for 2 h (Federici et al., 2014) and ⑥ 2 M NaOH at 55 °C overnight subsequently neutralization with 1 M HCl (Noordhuis et al., 2008).

The platinum lamp used in this work was operated at a current of 10 mA. Analysis was performed using an atomic absorption spectrometer (SpectrAA-240 FS; Varian, Palo Alto, CA, USA) to monitor the platinum absorbance at 265.9 nm corrected by deuterium background signals, with a slit width of 0.2 nm. Argon was used as the inert gas in all analyses. Standard solutions and samples were injected in duplicate on the platform inside the graphite tube by the auto-sampler needle. The furnace was programmed to execute a 75-s drying phase at 85–120 °C, followed by 8-s of ashing at 1,000 °C, then a 4.9-s atomization at 2,700 °C.

With 300 ug/L platinum solution as standard mother liquor, the blank and preparation solutions were the corresponding cell processing liquid mentioned above. The platinum standard addition solutions (60 ug/L, 120 ug/L, and 240 ug/L) used in this study were automatically generated by the instrument based on the mother liquor. Matrix effect was corrected on the pattern of standard addition. Calibration curves of different sample preparation methods were drawn according to the absorbance of standard addition solutions and correlation coefficients of all curves were calculated.

Accuracy and precision

Accuracy is defined as the agreement between the observed value and the true value and expressed as recovery rate in a percentage form. For the assessment of accuracy, high (125 ug/L), medium (85 ug/L) and low (45 ug/L) concentration standard solutions were added to corresponding high (110.9 ug/L, Hep G2), medium (96.5 ug/L, A2780) and low (48.6 ug/L, A2780) concentration samples (Shah et al., 1991; Westgard, 1981). Then, the three mixtures of standard solutions and samples were analyzed for six times. The average recovery rate was calculated as following: (average value of sextuple tests—sample concentration)/ concentration of standard solution. Generally, 90 to 100% recovery was considered acceptable (Westgard, 1981).

Precision, defined as the agreement of replicate measurements of the same sample and calculated as the relative standard deviation (RSD) (Lewen, 2011; Shah et al., 1991; Vouillamoz-Lorenz et al., 2001), was evaluated by performing six replicate injections at four different concentration levels of A2780 cells (the average concentration of Low level: 44.6 ug/L, Medium-low: 82.9 ug/L, Medium-high: 119.5 ug/L, High: 190.2 ug/L) (Tiwari & Tiwari, 2010). An analytical method was considered precise when the RSD was less than 15% except at the detection limit where 20% RSD was accepted (Shah et al., 1991; Tiwari & Tiwari, 2010).

Sensitivity

The sensitivity of an assay was evaluated by its detection limit, which is numerically equal to 3 times the standard deviation of at least 10 distinct measurements of blank samples (Vouillamoz-Lorenz et al., 2001).

Interference

Interference is defined as the effect of a component on the accuracy of the measurement of another component, which may result in high values (enhancement) or low values (inhibition) (Westgard, 1981). In this study, the interference from the metals coexisting in cells to platinum measurement was evaluated. Based on the content of metals contained in cells, a final concentration of 90.9 mg/L Ca2+ and Mg2+were added to 95.3 ug/L Hep G2 cell samples, 12.2 mg/L Zn2+, K+, Na+ and 1.2 mg/L Cu2+ and Mn2+ to 102.2 ug/L Hep G2 cell samples and 1.1 mg/L Fe2+ to a 75 ug/L Hep G2 cell sample. The platinum concentration in each mixture was measured for six times and the average value was used for calculation of the recovery rate, which was determined by the ratio of the platinum concentration in the sample with metal addition to that in the corresponding parent samples. In general, an average recovery rate of 90% to 110% was considered to be acceptable.

Stability

To evaluate the long-term storage stability, A2780, HeLa, and Hep G2 cells were stored at −20 °C and −80 °C for 7, 14, 21, 35 and 60 days before treatment with 0.2% HNO3 and GFAAS assays, and the platinum contents in these samples were compared with freshly harvested cell.

To assess the stability of platinum concentration during sample processing, the processed cell samples were stored in room temperature without sealing for three hours followed by GFAAS analysis. In addition, the platinum contents of samples prepared and stored in sealed tubes were detected over the next 24 h and 48 h. Stability was calculated by comparing the platinum concentrations assessed at each time point to the corresponding freshly prepared samples. The results within 85%–115% was considered to be acceptable (Kloft et al., 1999).

Method application

A2780 and CAOV3 cells were treated with 20 uM cisplatin for 12, 24, 36, 48, and 72 h. The intracellular platinum contents were analyzed and platinum concentration–time curves were drawn. In addition, the intracellular platinum accumulation in A2780 was compared with that in its cisplatin-resistant subline A2780/CDDP after incubation with 20 uM cisplatin for 24 h. Each assay was performed in triplicate.

Statistical analysis

Data were expressed as mean ± standard deviation. Differences between groups were analyzed using student’s t-test. All tests were two-tailed and P-values of <0.05 were considered statistically significant.

Results

Comparison of different sample preparation methods

To determine the intracellular platinum concentration, cell pellets were processed with 0.2% HNO3, 0.2% Triton X-100, concentrated nitric acid and RIPA combined with concentrated nitric acid respectively. Generally, the linearity of the calibration curves was satisfactory with a correlation coefficient higher than 0.99 in all cases (Fig. 1A). The platinum concentrations in samples treated with 0.2% HNO3 or 0.2% Triton X-100 were significantly higher than that in samples treated with concentrated nitric acid alone or in combination with RIPA (Fig. 1B), suggesting a significant loss of platinum caused by the processing with concentrated nitric acid. The spectrogram of samples processed using NaOH combined with HCl was undulate and lacked a wave crest at 265.9 nm. Hence, this method was excluded.

Figure 1 Comparison of different sample preparation methods.

(A) Calibration curves of different sample preparation methods according to the absorbance of a series of platinum standard addition solutions ranging from 60 µg/L to 240 µg/L. (B) Platinum contents of A2780, HeLa, Hep G2 cells processed using different methods. Student’s t-test, * P < 0.05, *** P < 0.001, # P > 0.05. (C) Platinum concentration of aliquots prepared using various concentrations of nitric acid.

As shown in Fig. 1B, the difference of platinum contents between samples processed with 0.2% HNO3 and 0.2% Triton X-100 was not statistically significant. Nevertheless, the treatment with 0.2% HNO3 was superior to 0.2% Triton X-100 in light of repeatability indicated by a RSD of 3.6% versus 24.7%. In addition, the impact of nitric acid concentrations ranging from 0.05% to 5% on the platinum determination was investigated. As shown in Fig. 1C, there was no significant difference in the platinum concentrations between groups. However, samples prepared with 0.2% HNO3 gave the littlest RSD among the different concentrations of nitric acid, suggesting a better ability of 0.2% HNO3 to eliminate matrix interference. Thus, the 0.2% HNO3 treatment was used to prepare cell samples for subsequent experiments.

Accuracy, precision and sensitivity

Given that the sample preparation method using 0.2% HNO3 was characterized with fewer platinum loss and lower RSD in the subsequent GFAAS assays, we thought to further evaluate its accuracy and precision. As shown in Table 1, the average recovery rates of samples at different platinum concentration levels ranged from 97.3% to 103.0%. The results summarized in Table 2 presented that the RSD were less than 5% (1.4%–3.8%) in all cases and tended to decrease with the increase of platinum content in samples. The detection limit of the assay analyzed was 13.23 ug/L.

Table 1 Evaluation of accuracy in terms of recovery rate at different concentration levels.

Cell lines	Concentration of sample (ug/L)	Added standard solution (ug/L)	Number of the experiment	Mean (ug/L)	Average recovery (%)	
			1 (ug/L)	2 (ug/L)	3 (ug/L)	4 (ug/L)	5 (ug/L)	6 (ug/L)			
A2780	48.6	45.0	95.5	85.9	92.5	92.2	93.3	95.0	92.4	97.3	
A2780	96.5	85.0	180.2	182.2	183.7	184.2	183.4	188.0	183.6	102.5	
Hep G2	110.9	125.0	232.9	235.7	237.0	241.8	242.8	247.5	239.6	103.0	

Table 2 Precision of the method evaluated as relative standard deviation at different concentration levels of A2780 cells.

Number of the experiment	Platinum concentration (ug/L)	
	Low	Medium-low	Medium-high	High	
1	42.8	80.4	117.1	186.3	
2	43.6	82.4	118.0	187.5	
3	43.9	82.5	119.9	190.6	
4	44.0	82.8	120.1	191.2	
5	45.8	83.4	120.4	191.8	
6	47.4	85.7	121.6	193.6	
Average	44.6	82.9	119.5	190.2	
RSDa (%)	3.8	2.1	1.4	1.5	
Notes.

a Relative standard deviation.

Interference

To evaluate the interference of the coexistent metals on this GFAAS-based platinum assay, eight metals including Ca, Mg, Zn, Cu, K, Na, Mn and Fe were added in samples containing platinum. As shown in Table 3, the average recoveries in all cases were between 95.1% and 103.1%, which indicated that platinum determination was not affected by additional  metal.

Table 3 Influence of coexistent metals on platinum determination in Hep G2 cell samples.

Metals	Added concentration (ug/L)	Sample Pta concentration (ug/L)	Number of the experiment	Mean	Average recovery (%)	
			1 (ug/L)	2 (ug/L)	3 (ug/L)	4 (ug/L)	5 (ug/L)	6 (ug/L)			
Ca	90.9	95.3	85.5	93.3	94.5	96.9	101.2	99.4	95.1	99.8	
Mg	90.9	95.3	104.0	96.5	100.8	94.0	93.5	99.3	98.0	102.9	
Fe	1.1	75.0	74.8	74.1	67.1	71.1	73.5	67.2	71.3	95.1	
Zn	12.2	102.2	111.5	95.4	92.6	103.2	95.3	98.1	99.4	97.2	
Cu	1.2	102.2	102.6	111.1	106.1	106.6	104.2	101.9	105.4	103.1	
Mn	1.2	102.2	96.0	95.8	108.9	107.5	107.6	107.6	103.9	101.7	
K	12.2	102.2	101.6	103.3	97.0	110.8	101.6	106.3	103.4	101.2	
Na	12.2	102.2	107.0	104.9	100.2	101.7	100.4	95.6	101.6	99.4	
Notes.

a Platinum.

Stability

As shown in Table 4, there was no significant decrease in intracellular platinum concentration during the 7–60 days storage at −20 °C and −80 °C, which was indicated by the recovery rates between 87.0% and 113.9% (Table 4).

Table 4 Long-term storage stability of platinum concentration in cell samples.

Cell lines	Storage conditions	0d (ug/L)	Storage time	
			7d	14d	21d	35d	60d	
			Ca (ug/L)	Average recovery (%)	Ca (ug/L)	Average recovery (%)	Ca (ug/L)	Average recovery (%)	Ca (ug/L)	Average recovery (%)	Ca (ug/L)	Average recovery (%)	
A2780	−20°C	110.9	112.7	101.6	126.3	113.9	124.6	112.4	116.2	104.8	96.4	87.0	
	−80°C	110.9	100.7	90.8	111.9	100.9	126.2	113.8	121.7	109.8	98.0	88.4	
HeLa	−20°C	109.4	113.6	103.8	104.7	95.7	113.6	103.9	96.8	88.5	112.3	102.7	
	−80°C	109.4	111.2	101.6	121.8	111.3	123.6	113.0	108.7	99.4	112.6	103.0	
Hep G2	−20°C	113.4	113.7	100.3	113.9	100.4	117.2	103.3	112.1	98.9	101.9	89.9	
	−80°C	113.4	117.0	103.1	102.3	90.2	111.8	98.6	107.6	94.9	122.2	107.8	
Notes.

a Concentration.

Compared with freshly processed samples, the results in Table 5 revealed no significant alteration in platinum content of samples processed and stored without sealing for three hour, excluding the effects of solvent evaporation during measurement on the test results, which can lead to an increase in platinum concentration. Moreover, the platinum concentrations in processed samples that were stored in sealed tubes over a period of 48 h were also found to be acceptable, with recovery rates between 92.8% and 111.2% (Table 5).

Table 5 Stability of platinum concentration during cell sample processing.

Cell lines	Starting point	Unsealed	Sealed	
	Ca (ug/L)	3 h	24 h	48 h	
		Ca (ug/L)	Average recovery (%)	Ca (ug/L)	Average recovery (%)	Ca (ug/L)	Average recovery (%)	
Hep G2	117.0	110.8	94.7	108.6	92.8	111.1	95.0	
A2780	98.0	91.6	93.5	104.4	106.5	102.4	104.5	
A2780	40.0	44.5	111.2	43.3	108.2	42.2	105.5	
Notes.

a Concentration.

Method application

The proposed method was applied to determine the dynamic change of platinum accumulation in ovarian cancer cells. After incubation with cisplatin, the intracellular platinum concentrations in A2780 and CAOV3cells increased in a time-dependent manner form 0 h to 24 and 36 h respectively and gradually declined afterward (Fig. 2A). The platinum concentrations in A2780 and its resistant variant A2780/CDDP were also measured. Consistent with the literature reported (Zisowsky et al., 2007), the intracellular platinum accumulation was significantly decreased in the A2780/CDDP cells compared with the A2780 (P < 0.01, Fig. 2B).

Figure 2 Practical application of the method.

(A) The platinum concentration-time curves of A2780 and CAOV3 cells. (B) Comparison of platinum accumulation in A2780 and its resistant variant A2780/CDDP after incubation with 20 uM cisplatin for 24 h. Student’s t-test, ** P < 0.01.

Discussion

The determination of platinum concentration in cultured cells is useful in studying the mechanisms of platinum-resistance in tumor. Here, we identified a GFAAS-based assay for quantitative platinum detection with high accuracy and precision. We compared different cell treatment procedures and found 0.2% HNO3 treatment was optimal for subsequent GFAAS platinum analysis with less platinum loss and high repeatability, which might be partially attributed to its simplicity.

High stability of platinum concentrations in samples is a prerequisite for reliable quantification. Burger (Burger et al., 2010),Buss (Buss et al., 2011) and Zisowsky (Zisowsky et al., 2007) stored samples at −20 °C, while Takahashi (Takahashi et al., 1993) and Neill (O’Neill, Hunakova & Kelland, 1999) suggested to store samples at −80 °C or in liquid nitrogen. In the present study, for the GFAAS-based platinum detection, we found that cell samples were allowed to be stored at −20 °C or −80 °C for at least two months. In addition, we found the platinum concentrations in processed samples could remain stable at room temperature for at least three hours that ensures the comparability of samples with different waiting time before GFAAS analysis.

Internal standard method, standard curve method and standard addition method are often used to quantify target metals in a variety of sample types by GFAAS. For the success of an analysis, selecting a proper analytical method is important. The method of internal standardization could provide a compensation of analyte losses during sample preparation, minimize the instrumental variation and correct matrix effect (De Baere et al., 2012). Nevertheless, it is inconvenient to select a proper internal standard in many cases, for that the physical and chemical properties of an internal standard, such as emission wavelength, should be similar to that of the analyte and the internal standard must not be contained in the sample (Zheng et al., 2001). Through the standard curve method, the analytical signal is directly correlated with the concentration of analyte (Honorato et al., 2002), which mainly applies in the situation that the matrices of samples are close to that of the standard solutions. Obviously, standard curve method is not appropriate for platinum detection in cultured cells because of their complicated components. The standard addition method, whereby specific quantities of test samples are spiked with standard solutions, which can tremendously reduce the errors stemming from the physical and chemical differences between the matrices of samples and that of the standard addition solutions (Honorato et al., 2002). Thus, considering the ability to reduce the interference from proteins, nucleic acids and other components contained in cells, the standard addition method was used in the present work. Consistent with this effect, interference experiments revealed that metals at macro (K, Na, Ca, Mg) and trace levels (Fe, Cu, Mn, Zn) had no significant influence on the quantification of platinum.

Wills et al. reported the measurement of cadmium (Cd) levels in cultured retinal pigment epithelium cells using GFAAS and the results were compared with that detected by ICP-MS. They found that the two methods gave essentially identical results with ±5% standard deviation (Wills et al., 2008). Additionally, a study on the compatibility between ICP-MS and GFAAS for Cd detection based on 1,159 blood samples showed a close correlation between the results by the two methods and suggested both methods could be used inter-convertibly when Cd was >2 ug/L (Fukui et al., 2011). Although the determination limit of ICP-MS is substantially lower than that of GFAAS, both are sufficient for detection of metals such as Cd in human blood (Fukui et al., 2011). All together, these findings confirmed the accuracy of GFAAS.

Reduced drug accumulation in tumor cells has been a well-known mechanism of platinum resistance. Quantifying intracellular platinum content using GFAAS-based assay might promote the research of platinum resistance. Burger et al. (2010) proved that organic cation transporter 2, which is involved in the translocation of endogenous and exogenous compounds across epithelial membranes, was also implicated in the cellular uptake of platinum-containing anticancer drugs and contributed to the development of resistance. In addition, the technique of GFAAS can be applied to evaluate the effect of drug modification and combination therapy. Enhancing the lipophilicity of oxaliplatin analogues by introducing carrier ligands could promote the early influx rate of drugs and increase intracellular platinum accumulation subsequently (Buss et al., 2011). The use of macromolecular prodrugs (e.g., platinum-albumin complex and platinum-polyethylene glycol complex) exploiting endocytosis as alternative uptake mechanism also has the potential to increase cellular platinum accumulation and overcome multidrug resistance (Garmann et al., 2008). The combination of cisplatin and two phytochemicals (curcumin and epigallocatechin-3-gallate) could produce synergistic outcomes, i.e., greater cellular accumulation of platinum (Yunos et al., 2011). Recently, GFAAS has also been applied to investigate the newly synthesized platinum-nitroxyl complexes and platinum (IV) texaphyrin conjugate for their potential to circumvent cisplatin resistance (Cetraz et al., 2016; Thiabaud et al., 2014), which indicates that quantifying intracellular platinum concentration using the technique of GFAAS might play a role in new drug research and development.

Conclusions

A GFAAS-based assay following sample preparation using 0.2% nitric acid for determination of platinum accumulation in cultured cells was validated in this work, which was proved to be accurate, sensitive, simple and cost-effective and might improve the research of platinum-based antitumor therapy.

Supplemental Information

Figure S1 The absorbance of standard addition solutions for different sample preparation methods

Click here for additional data file.

Figure S2 Platinum concentrations of the aliquots of A2780, HeLa, HepG2 cells processed using different methods

Click here for additional data file.

Figure S3 Platinum concentration of aliquots prepared using various concentrations of nitric acid

Click here for additional data file.

Figure S4 Platinum concentrations of A2780, CAOV3 and ACP cells incubated with cisplatin

A2780 and CAOV3 cells were treated with 20 uM cisplatin for 12, 24, 36, 48, and 72 hours and the intracellular platinum contents were analyzed. The cisplatin-resistant subline A2780/CDDP was incubated with 20 uM cisplatin for 24 hours.

Click here for additional data file.

Table S1 The platinum concentrations of samples stored under different conditions

Click here for additional data file.

Additional Information and Declarations

Competing Interests

Author Contributions

Human Ethics

Data Availability

The authors declare there are no competing interests.

Man Xiao conceived and designed the experiments, performed the experiments, analyzed the data, wrote the paper, prepared figures and/or tables.

Zaiju Huang performed the experiments, contributed reagents/materials/analysis tools.

Jing Cai conceived and designed the experiments, analyzed the data, wrote the paper, reviewed drafts of the paper.

Jinghui Jia performed the experiments, analyzed the data, prepared figures and/or tables.

Yuzeng Zhang performed the experiments, analyzed the data, contributed reagents/materials/analysis tools.

Weihong Dong contributed reagents/materials/analysis tools, prepared figures and/or tables.

Zehua Wang conceived and designed the experiments, reviewed drafts of the paper.

The following information was supplied relating to ethical approvals (i.e., approving body and any reference numbers):

The present study was approved by the ethical committee of Union Hospital, Tongji Medical College, Huazhong University of Science and Technology, China (Approval number: 2014073).

The following information was supplied regarding data availability:

The raw data has been supplied as a Supplemental File.

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
