# Peer review of "Comparison of different sample preparation methods for platinum determination in cultured cells by graphite furnace atomic absorption spectrometry"

_PeerJ, doi:10.7717/peerj.2873_

## Round 0.1 · original submission · Minor Revisions

· Academic Editor

Minor Revisions

The paper is focused on a new sensitive assay to quantify the intracellular platinum content that may be helpful during platinum-based therapy of human cancer. Most of the comments by the Reviewers do not require further experiments, but they just indicate some technical/procedural aspects to be clarified. Please, introduce all the required information and provide an accurate point-by-point response to the reviewers criticisms.

Reviewer 1 ·

Basic reporting

The manuscript “Comparison of different sample preparation methods for platinum determination in cultured cells by graphite furnace atomic absorption spectrometry” by Xiao M et al. demonstrated that a GFAAS-based assay was an accurate and sensitive method to quantify the intracellular platinum content and may be suitable for the assessment of platinum-based therapy of cancer cells.
In the present article, the authors have not always used a clear and unambiguous text. There are some comments:
-In the “Methods” section of the Abstract the authors should add more information referred to cell line types used in the study and more details referred to sample preparation before the analysis for platinum determination.
- In the Materials and Methods section of the article at paragraphs “ Cell lines and cell culture” and “Sample preparation and operating condition of instrument” is not clear if all cell suspension aliquots were frozen before the analysis or if only the samples that were studied for long term storage stability of platinum concentration were frozen for different time (Table 4). In this section the authors should better describe these aspects of the sample preparation.

Experimental design

The present study addresses a research questions within scope of the journal. The research question is well defined but there are some comments:
In the “Results” section at the paragraph “Comparison of different sample preparation” the authors compared the different methods to determine the intracellular platinum concentration. In particular, when they described the results summarized in the figure 1B, they concluded that the highest platinum concentration were found in samples treated with 0.2% HNO3 and 0.2% Triton X-100. Moreover, it is indicated that the treatment with 0.2% HNO3 was superior than 0.2% Triton X-100 due to a low relative standard deviation of the reported data set. To support this conclusion, the authors should also indicate in the text and in the figure 1B, if there is a statistically significant difference between cell treatment with 0.2% HNO3 and 0.2% Triton X-100 especially for A2780 and Hela cell lines.

Validity of the findings

- In the “Discussion” the authors reported that the quantitative platinum detection in cultured cells with high accuracy and precision is a useful approach to study the mechanisms of platinum resistance in tumor. To this respect, they should better describe the translational relevance of the study to improve cancer therapy.

Reviewer 2 ·

Basic reporting

The manuscript, "Comparison of different sample preparation methods for platinum determination in cultured cells by graphite furnace atomic absorption spectrometry” is well written and the target is set up clearly.

Experimental design

Figure 1C. The authors declare no significant difference in the platinum concentrations between groups following HNO3 treatment, but the standard deviations are substantial. However, they selected the concentration that gives the littlest standard deviation. The authors should specify if this choice is due to technical problem.
Figure 2. The authors should explain why Hela and HepG2 cell lines were not included in the analysis.
Figure 2B. The authors need to specify the experimental timing
Table 1, 2, 3 and 5. The authors should specify the cell lines in which the analiys were performed.

Validity of the findings

A large number of patients who receive anticancer chemotherapy are efficiently treated with Platin-based anticancer drugs. Indeed an intensive research, which examines the role and the amount of platinum in biological systems, has been completed. This effort enriches the literature of a technical clarification that is useful to improve the quantitative analysis of platinum in cellular systems

Additional comments

In a precise way, the authors emphasized the importance to set a method that allows to determine changes in the chemotherapeutics platimun based agents accumulation in cancer cells with high efficiency. To address toward this scope, they compared different cell treatment procedures to quantify the intracellular platinum amount using GFAAS assay. They evaluated the accurancy and precision of the methods, the interference of others metals and the platinum stability in different cancer cell lines. The authors describe in clear and reproducible way the procedure.

---

## Round 0.2 · accepted · Accept

· Academic Editor

Accept

The authors responded in a clear and satisfactory way to the referees comments. Therefore, I recommend to accept the manuscript "Comparison of different sample preparation methods for platinum determination in cultured cells by graphite furnace atomic absorption spectrometry".

Reviewer 1 ·

Basic reporting

No comments

Experimental design

No comments

Validity of the findings

No comments

Additional comments

The authors have addressed the comments and suggestions provided in the previous review. There are no further comments

Reviewer 2 ·

Basic reporting

"No Comments"

Experimental design

"No Comments"

Validity of the findings

"No Comments"

Additional comments

In my opinion the authors’ responses to the comments of reviewers have been clear and satisfactory. I recommend to accept the manuscript "Comparison of different sample preparation methods for platinum determination in cultured cells by graphite furnace atomic absorption spectrometry"